# Problems Encountered Using Fungal Extracts as Test Solutions for Fungal Allergy Diagnosis

**DOI:** 10.3390/jof9100957

**Published:** 2023-09-23

**Authors:** Sandra Pfeiffer, Ines Swoboda

**Affiliations:** The Molecular Biotechnology Section, Department Applied Life Sciences, FH Campus Wien, University of Applied Sciences, 1100 Vienna, Austria; sandra.pfeiffer@fh-campuswien.ac.at

**Keywords:** fungal allergy, fungal allergen extracts, fungal allergens, allergy diagnosis, fungal cultivation

## Abstract

Fungal allergy is a worldwide public health burden, and problems associated with a reliable allergy diagnosis are far from being solved. Especially, the lack of high-quality standardized fungal extracts contributes to the underdiagnosis of fungal allergy. Compared to the manufacturing processes of extracts from other allergen sources, the processes used to manufacture extracts from fungi show the highest variability. The reasons for the high variability are manifold as the starting material, the growth conditions, the protein extraction methods, and the storage conditions all have an influence on the presence and quantity of individual allergens. Despite the vast variety of studies that have analyzed the impact of the different production steps on the allergenicity of fungal allergen extracts, much remains unknown. This review points to the need for further research in the field of fungal allergology, for standardization and for generally accepted guidelines on the preparation of fungal allergen extracts. In particular, the standardization of fungal extracts has been and will continue to be difficult, but it will be crucial for improving allergy diagnosis and therapy.

## 1. Introduction

Already at the beginning of the 18th century, the exposure to fungal spores and fragments was recognized as a cause of adverse respiratory symptoms [1]. In 1726, it was reported that a patient experienced a severe asthma attack after visiting a wine cellar, where must was fermented [2]. Nevertheless, the association between exposure to fungi and the occurrence of allergic symptoms has been discussed controversially for a long time, and even today, fungi are still a neglected allergen source [3,4]. However, nowadays, data from several epidemiological studies provide evidence for the important role of fungi in respiratory disease [3,4].

Fungi, together with mites and pollen, represent the three main causes of respiratory allergy that induce allergic diseases such as rhinitis, asthma, allergic bronchopulmonary mycoses, and hypersensitivity pneumonitis [4,5,6]. Fungi are principally dispersed as airborne spores, but also fungal hyphae and fragmented mycelia are aerosolized in large numbers [7] and allergens produced by fungi can occur in both, the spores as well as in the mycelium [8,9,10,11,12,13]. In contrast to other respiratory allergen sources, exposure to fungi occurs universally, outdoors and indoors, including within occupational settings, and is therefore impossible to avoid [14,15,16,17,18].

### 1.1. Prevalence of Fungal Sensitization

Fungal allergy is a common worldwide health problem [19,20,21]. The prevalence of fungal allergy is estimated to range from 3% to 10% in the general population and up to 44% of atopics and 80% of asthmatics are sensitized to at least one fungal species [3,4,16,19,22,23,24,25]. Moreover, Rodriguez-Orozco et al. (2010) and Forkel et al. (2021) documented an alarming increase in sensitization to fungi [26,27].

However, the precise prevalence of sensitization to fungal allergens remains unclear due to the high variability between different studies [3,4,19,23,28]. Possible explanations for the variation in sensitization to fungi are the patient populations studied as well as the variable level of fungal exposure [20,25,29,30]. In addition, due to the lack of standardization, fungal extracts of high variability and poor quality are commonly used as test solutions for allergy diagnosis [20,23]. Moreover, so far, only a few fungal species have been extensively studied concerning their allergenic potential, leading to a general lack of knowledge about fungal species that induce allergic reactions [3,4,12,19,23,28]. Consequently, it is assumed that the sensitization rates to fungi tend to be underestimated [3,4,12,19,23,28].

### 1.2. Allergenic Fungal Species

It is estimated that at least 5 to 6 million fungal species exist worldwide, of which approximately 100,000 species have been described [5,31]. Eighty fungal genera have so far been shown to induce type I allergic reactions in atopic individuals [20,22,23,32,33]. However, only a few species have been extensively investigated as causative agents of hypersensitivity reactions [12]. *Alternaria*, *Cladosporium*, *Aspergillus*, *Penicillium*, *Fusarium*, *Epicoccum* and *Curvularia* are classically considered the most important fungal genera that cause allergic diseases [3,4,16,21,25,27,30,34]. Nevertheless, given the large taxonomic diversity within the fungal kingdom and the results of different aerobiological studies, it is very likely that various other fungal genera and species also have to be considered as allergen sources [7,16,34].

### 1.3. Fungal Allergens

Fungi are regarded as complex allergen sources as their allergenic molecules can be found in the spores as well as in the mycelia [3,4]. Furthermore, many fungi are known to produce a wide range of allergens that can either be specific for a certain fungal species, genus, or family, or can occur in different closely and distantly related fungal species [3,4]. Moreover, a great number of homologous fungal allergens are cross-reactive [3,4]. To date, 120 proteins from 31 different fungal species have been officially recognized as allergens by the World Health Organization and International Union of Immunological Societies (WHO/IUIS) Allergen Nomenclature Sub-committee [35]. Of the 31 allergenic species listed, most allergens have so far been identified in *Alternaria alternata* (12 allergens) and *Aspergillus fumigatus* (30 allergens) [35].

As mentioned, fungi produce complex repertoires of allergens that are specific for a certain species (e.g., Asp f 2), genus (e.g., Asp f 1), or family (e.g., Alt a 1) [36,37,38,39,40,41]. For example, the major fungal allergen Asp f 1 is considered as a genus-specific allergen, with homologous, cross-reactive molecules occurring in other species of the genus *Aspergillus* [36,37].On the other hand, the major allergen Alt a 1 from the well-studied fungal allergen source *A. alternata* represents a family-specific allergen as homologous allergens have been reported in different species of *Alternaria* and in closely related species, including *Stachybotrys botryosum* and *Ulocladium chartarum* [38,39,40,41], which belong to the same *Pleosporaceae* family. However, allergens cross-reactive with Alt a 1 are not present in more distantly related fungal allergen sources such as the genera *Cladosporium* and *Aspergillus* [39]. Thus, *A. alternata* shows limited cross-reactivity with the allergenic fungi *A. fumigatus*, *Penicillium chrysogenum*, and *Cladosporium cladosporioides*, but has significant levels of allergenic cross-reactivity with other fungi belonging to the *Pleosporaceae* family [39].

So far identified cross-reactive fungal allergens belong to different protein families, including proteolytic enzymes, stress response proteins, proteins involved in protein synthesis, and proteins involved in carbohydrate metabolism [32]. The enzyme enolase represents a prominent example of a highly conserved, cross-reactive allergen family that has been identified in various fungal species [41,42,43,44,45,46].

The current review addresses problems and obstacles that are encountered when using fungal extracts as test solutions for fungal allergy diagnosis.

## 2. Diagnosis of Fungal Allergy

Similarly to the diagnosis of other allergies, the diagnosis of fungal allergies is also a stepwise process, including anamnesis, the determination of total and allergen-specific IgE antibodies, and skin prick tests (SPT) [3,22,47]. The accuracy and reliability of any in vivo and in vitro assays is highly dependent on the quality of the material used for testing [21,22,23]. Especially in case of fungal allergy, the lack of readily available high-quality extracts is a major problem for diagnosis [21,47].

### 2.1. Variability of Commercial Fungal Extracts

Several studies have shown that the accuracy and correlation of the results from in vivo and in vitro assays are often not in concordance due to the variability of the fungal extracts used in the different diagnostic tests [43,48,49,50,51,52,53,54,55,56,57,58,59,60].

For example, Yunginger et al. (1976) compared the relative in vivo and in vitro allergenic potencies of 12 commercially available *Alternaria* extracts and found considerable differences in their reactivity [49]. Vijay et al. (1984) studied batch-to-batch variations in *Alternaria* extracts from two companies and revealed, besides differences in their biochemical composition, a 40-fold variation in potency in batch-to-batch products from the same company and a 55-fold difference between extracts from the two companies [50]. A review published by Esch et al. (2004) describes that *A. alternata* extracts from different manufacturers significantly differed in their carbohydrate, protein, and Alt a 1 content and therefore also in their relative allergenic potency [51]. When adjusted to equivalent weight/volume concentrations, the protein and carbohydrate contents varied by more than 100-fold [51]. Also, the compositional profiles, evaluated by means of SDS-PAGE, of various *A. alternata* products manufactured by different companies were highly variable [51,52]. Furthermore, Twaroch et al. (2016) revealed that commercial *Alternaria* skin-prick solutions contain varying amounts of Alt a 1 and lack other *Alternaria* allergens [53]. In addition, Unger et al. (1999) showed that commercially available *A. alternata* extracts failed in 2 out of 10 cases to correctly diagnose allergic patients and also led to false-positive results in 2 of 10 control individuals [43].

With the use of 26 *Aspergillus*-reactive human IgE mAbs, directed against epitopes of different allergens, Wurth et al. (2018) demonstrated the tremendous variability of relevant allergens in commercial *Aspergillus* extracts, as only four of the antibodies recognized allergens in all the extracts [54]. The absence of secreted proteins, such as Asp f 1, and glycanosyltransferase in all the tested extracts except one suggests that some of the observed heterogeneity is likely due to the source material used to prepare the extract (i.e., mycelium, spores or growth medium) [54]. This study further raises significant concerns about the reliability of the diagnostic and therapeutic use of the currently available commercial *Aspergillus* allergen extracts [54].

Similar variations in the allergen, protein and carbohydrate contents and batch-to-batch variations in commercially available extracts were also described for fungal extracts of several other fungal species, namely *Cladosporium herbarum*, *Penicillium notatum*, *P. chrysogenum*, *Mucor racemosus*, *A. fumigatus*, *Epicoccum nigrum*, *Helminthosporium sativum* and *Aureobasidium pullulans* [51,52,54,55,56,57,58,59,60].

### 2.2. Fungal Extracts–Imperfect but Not Yet Obsolete

Even though several studies have demonstrated the poor quality and high variability of fungal allergen extracts, they still form the basis of all diagnostic tests [20,22]. They are used for diagnosis, despite the fact that none of the extracts available has been standardized and approved by the European Medicines Agency (EMA) or the Food and Drug Administration (FDA) [3,12,23,61]. The other serious problem is that allergen extracts for diagnosis are currently only available of a limited number of fungal species (summarized by Esch et al. (2004) [51] and by Linneman et al. (2016) [62], whereas for most allergenic fungal species, there are no allergen extracts available [3]. This is certainly another important factor that contributes to the underdiagnosis of fungal allergy [3,51].

## 3. Preparation of Fungal Allergen Extracts

Compared to the manufacturing processes used for the production of allergen extracts from other allergen sources such as pollen, animal dander, mites, insects, hymenopteran venoms and foods, the processes used to manufacture extracts from fungi, as depicted in Figure 1, show the highest variability [12,23,63]. First, the raw material and fungal cultivation are important [23,51,63]. This is dependent on the choice of the specific fungal strain and the growth conditions, such as the culture medium, type of culture (i.e., submerge or plated cultures), temperature, exposure to light, sporulation and the duration of cultivation [51]. The second step of fungal extract production—the extraction procedure—is also a source of variation [23,52,63]. The extraction methods, extraction media and additives can influence the final protein, antigen and allergen content of the extract [23,52,63]. Moreover, these factors are highly dependent on whether fungal spores, mycelia or the growth medium is used for the extract preparation [23,52,63]. Afterwards, as not only allergens, but also non-allergenic molecules are extracted from the fungal material, proteins need to be purified from the crude extracts. Quality control, the final step in the procedure, is another potential source of variability, where the allergen quantification methods and the used standards are likely sources of variability [23,52,63].

### 3.1. Source Material and Strain Variability

The preparation of allergenic extracts begins with the selection of the source material, and the quality and purity of the source material are the basis for the production of high-quality extracts [64,65]. In case of fungal extracts, the selection of the source material already poses the first problem, as fungi have the tendency to mutate frequently [66]. This leads to the generation of new fungal strains, but also to intra-strain variabilities [67,68,69,70]. The high rate of somatic mutations is one mechanism by which fungi, as decay organisms, can rapidly adapt to changes in available food sources [66,71]. Such mutations that frequently occur in nature cause differences between strains, but also within a single strain, that can affect the protein composition and the allergenic potency of protein extracts [10,49,60,67,69,70,72,73,74,75].

Variability between different strains and among single strains has been extensively studied for *A. alternata* in terms of allergological or immunological and biochemical characteristics [10,49,50,53,67,69,70,74]. Portnoy et al. (1993) showed that strain differences do qualitatively and quantitatively affect the contents of *Alternaria* preparations [10]. These conclusions were reinforced by the study of Twaroch et al. (2016), who found considerable variation in the reaction of IgE antibodies from allergic patients to extracts derived from four different *Alternaria* strains [53]. Distinct biochemical differences (e.g., content of nitrogen and carbohydrate) were further seen between extracts prepared from different strains of the same *A. alternata* species by Schumacher et al. (1976) [69].

The strain-dependent variation in the allergenic potential of *A. alternata* extracts was also demonstrated using the species’ major allergen, Alt a 1, as marker. For example, Martinez et al. (2006) revealed a high variability of Alt a 1 expression in different strains of *A. alternata*, with coefficients of variation of more than 130% [67]. Other studies showed that some strains of *A. alternata* produce significant quantities of the major allergen Alt a 1 only under certain conditions, and some strains do not produce detectable quantities under any condition [76]. Rosenthal et al. (1998) suggested that the variability in the Alt a 1 allergen content in *A. alternata* extracts was the result of posttranslational events, as they found comparable amounts of Alt a 1 encoding mRNAs in different *A. alternata* strains [38].

These data indicate the great variability between, and sometimes also within, fungal strains and illustrate how strain-dependent differences can influence the quality of fungal allergenic extracts [12,64,67,77]. Therefore, it is of utmost importance that the material used to produce fungal extracts should be derived from well-defined, pure cultures from established suppliers such as the American Type Culture Collection (Manassas, Virginia) or the Centraalbureau voor Schimmelcultures (Utrecht, the Netherlands). In addition, fungi should not be kept in culture for a long time due to the increasing risk of genetic mutations [10,12,67].

### 3.2. Fungal Cultivation

Different fungal species and, as mentioned above, sometimes even different strains of a certain species, require different growth conditions for optimal fungal growth [78,79]. Since each manufacturer uses its own culture medium and own culture conditions, qualitative differences between fungal allergen extracts produced by different companies are expected, even when identical strains of fungal stock cultures are used [12,51,54].

It is known that variations in the nutritional composition of the culture medium have an influence on the morphologic, biochemical, and allergenic characteristics of a given fungal strain [12,51,79]. Furthermore, other culture parameters, such as the temperature, the exposure to light and the growth time, have also been shown to affect the allergenic composition of an individual fungal species or strain [51,78,80,81]. Therefore, the knowledge of the culture conditions that lead to the highest allergen expression is the first step toward the standardization of raw materials for fungal allergen extracts [78,81,82,83]. However, the optimization of the growth parameters for the maximum expression of allergens is not easily achieved, because we and others described that different allergens might be produced at different stages of the fungal growth cycle and that they might require different growth conditions for the expression of highest amounts [78,81,83].

#### 3.2.1. Cultivation Medium

Different studies have shown that the type of medium (i.e., liquid or solid medium) as well as the medium composition used for fungal cultivation can play an important role for the quality and quantity of fungal raw material obtained for the preparation of fungal extracts [52,78,83,84,85,86,87,88,89].

The type of medium, whether liquid or solid, used for fungal cultivation can have an impact on fungal allergen expression. For example, cultures of *Curvularia lunata*, grown on potato dextrose agar (PDA), a solid fungal growth medium, resulted in the maximum yield of fungal raw material and the highest protein content in the extracts, whereas the growth in liquid Saborauds’ broth (SB) medium, a comparable fungal growth medium containing the same carbon source, led to the production of extracts with the highest allergenic potency [78]. Liquid SB medium was also shown to be ideal for the cultivation of *E. nigrum*, as it resulted in the highest amounts of allergenic proteins in the extracts [81]; however, this was only the case when yeast extract was added to the medium [81]. Nevertheless, the growth of *E. nigrum* on solid media, such as PDA, also resulted in potent allergen extracts [81]. The study by Ibarrola et al. (2004) reinforced the conclusion that liquid media encourage fungal allergen expression, as they detected a 15-fold stronger IgE-binding activity to fungal extracts prepared from *A. alternata* grown in liquid culture compared to agar-cultivated material [84]. In addition, Little et al. (1993) showed that agitated culture promoted submerged somatic mycelial growth, while static culture promoted surface sporulation and pigment production [85]. Moreover, the final mycelial mass was greater in agitated cultures than in static cultures after 28 days of incubation [81]. Interestingly, it was seen that the number of allergens secreted into the growth medium was much higher in static cultures as compared to agitated cultures [81].

In addition, individual components of the cultivation medium, such as the carbon source, can have an impact on fungal growth and therefore also potentially on fungal allergen expression [83,86,87]. In the study presented by Flaherty et al. (1970), *Trichophyton rubrum* was cultivated using different concentrations of glucose to analyze the influence of the carbon source on the allergenicity of fungal extracts prepared from mycelium [86]. The results indicated that higher carbon source concentrations in the medium (e.g., glucose) result in higher amounts of total nitrogen, soluble proteins, total carbohydrates and soluble carbohydrates in the allergenic extracts [86]. Similar effects were shown in the study by Kauffman et al. (1980) [87], where higher glucose concentrations in the medium resulted in higher yields of *A. fumigatus* mycelium mass [87].

Furthermore, not only the concentration, but also the type of carbon source can play a role in fungal allergen expression [83,88,89]. McAlister et al. (1981) and Machida et al. (1996) showed that the expression of allergenic enolases from *Aspergillus oryzae* and *Saccharomyces cerevisiae* was highly dependent on the carbon source added to the medium [88,89]. In our previous study we so that also the expression of the enolase from *U. chartarum*, Ulo c 6, depended on the carbon source as the growth on more complex carbon sources, such as cellulose and methylcellulose, led to an earlier onset of expression of Ulo c 6 as compared to the growth on the easily accessible substrate glucose [83]. However, this effect seems to be allergen specific as the expression of the major fungal allergens Alt a 1 and Ulo c 1 from *A. alternata* to *U. chartarum* was not affected by the type of carbon source present in the growth medium [83].

To conclude, it is important to clearly define the type of medium and its composition used for fungal cultivation for a consistent production of allergens. The problem is that a carbon source might have an impact on the expression of one allergen of a certain fungal species, whereas other allergen of the same species might not be affected [83]. In addition, even though previous studies have shown that the cultivation in liquid medium tends to lead to the production of allergen extracts of higher IgE reactivity, the cultivation of fungi on solid medium mimics more closely the in vivo situation and leads to a better fungal growth [78,81,84].

#### 3.2.2. Cultivation Temperature

Temperature is the most important physical environmental factor regulating the growth and reproduction of fungi, and each fungal species has its optimal growth temperature [90]. In addition, the impact of the growth temperature on fungal allergen expression has also already been investigated for several fungal genera, such as *Aspergillus*, *Alternaria* and *Ulocladium* [83,84,85,90].

Little et al. (1993) analyzed the effect of the cultivation temperature on the growth and allergenicity of *A. fumigatus* grown in liquid medium and showed that the growth at 25 °C and 37 °C led to an initial lag period, followed by an increase in mycelial weight [85]. Even though the initial growth rate was higher at 37 °C than at 25 °C, the final mycelial yield was higher at the lower incubation temperature [85]. However, a significantly higher number of allergens were observed in the growth medium during incubation at 37 °C than at 25 °C [85]. This increase in allergenicity found in the growth medium at 37 °C could either be due to an increase in hyphal lysis or in allergen production [85]. 

Hubballi et al. (2010) studied the effect of the cultivation temperature on the mycelial growth of *A. alternata* and showed that *A. alternata* grew better at 30 °C, than at 25 °C or 35 °C, and the lowest growth was seen at 5 °C [90]. It was further shown that the amount of the species’ major allergen Alt a 1 doubled at a growing temperature of 25 °C compared to 20 °C [90]. Thus, a cultivation temperature of 25–30 °C seems to be ideal for the growth of *A. alternata*, especially to produce extracts containing Alt a 1 [90]. Our own experiments, where we analyzed the impact of different cultivation temperatures, namely 20–25 °C, 30 °C and 37 °C, on the growth of *A. alternata* and *U. chartarum* and on the expression of the species’ major allergens Alt a 1 and Ulo c 1, showed that even though the cultivation temperature can have an impact on fungal growth, it does not necessarily influence fungal allergen expression as the major allergens were constantly expressed under each condition tested and could still be extracted from very little fungal biomass [83]. A different effect was seen for two other allergens: the species’ allergenic enolases [83]. While Alt a 6, the enolase from *A. alternata,* was not expressed under any condition, the enolase from *U. chartarum* was only expressed when grown at 37 °C for 20 days, suggesting the impact of the growth temperature on the expression of fungal allergens [83].

These results illustrate the impact of the cultivation temperature not only on fungal growth, but also on the expression of some allergens. The results also show that these effects do not always correlate. A cultivation temperature ideal for fungal growth can at the same time inhibit the expression of certain allergens.

#### 3.2.3. Exposure to Light

The exposure to light is known to have an impact on fungal growth and has been suspected to also play a role in fungal allergen expression [90]. Hubballi et al. (2010) studied the effect of light on mycelial growth of *A. alternata* [90]. The species was exposed to either continuous light, continuous dark or to 12 h light followed by 12 h dark periods [90]. Light increased the mycelial growth of *A. alternata*, but alternate cycles of 12 h light and 12 h darkness for 10 days resulted in the maximum mycelial growth [90]. However, the study did not analyze the effect of the exposure to light on allergen expression [90].

The effect of the exposure to sunlight has specifically been studied in the context of spore viability. Spores are specialized cells that are produced by fungi for reproduction, dispersal, and survival [7]. Airborne spores are the main dispersal units of fungi [7]. Therefore, spores, alongside fragmented mycelia, are also the cells with which allergic individuals get into contact with and which are regarded as the main source of fungal allergens [7]. In nature, the exposure to sunlight has a strong effect on the viability of airborne spores [91,92,93,94]. It was, for example, shown by Mitakakis et al. (2003) that a prolonged exposure to sunlight reduced the metabolic activity and germinability of *A. alternata* spores [91]. This was in accordance with results from Rotem et al. (1995), who showed that solar radiation directly reduced the survival of *Alternaria* spores [92]. Mitakakis et al. (2003) further explored whether the exposure to sunlight also affected the allergen releasing capacity of spores [91]. They saw that despite reducing the metabolic activity and germinability of spores, sunlight had no severe effect on the allergen releasing capacity of the spores [91].

In summary, whereas light and sunlight appear to have a positive impact on fungal growth and a negative effect on spore variability, no direct effect on fungal allergen expression seems to exist.

#### 3.2.4. Growth Time

It has been shown that the ideal growth period of fungi to produce allergen extracts is highly species-specific, but usually 2–4 weeks of growth are optimal, because after this time most species have produced spores and sufficient amounts of mycelial material [95]. The impact of the cultivation time on the expression of fungal allergens and on their presence in the produced protein extracts has been extensively studied for *A. fumigatus* [9,82,87,96].

For instance, Kauffman et al. (1985) studied the impact of the growth time on the growth of *A. fumigatus* and on the release of the species’ allergens into the culture medium [96]. It was shown that the production of secreted allergens was not a continuous process, but was indeed dependent on the state of growth [96]. During their first growth phase (phase 1, 0–6 days), characterized by a decrease in the pH of the growth medium (from 5.5 to <5.0) and by an increase in the mycelium weight, a limited number of allergenic components was produced [96]. During a second phase (6–16 days) after the pH had increased again to 8.0–8.5, the maximum mycelium yield was obtained after 10 days of cultivation and the second period of the liberation of allergens started [96]. A latter release of allergens in a third phase was mainly the result of lytic processes and the liberation of proteins after cell death, as was concluded from the decline in mycelium weight [96]. Thus, it was suggested that further cultivation can subsequently lead to a decline in the extract’s allergenic properties due to proteolysis and degradation [96].

These results were in accordance with studies carried out by van der Heide et al. (1985) and Gupta et al. (1999) [78,81]. Van der Heide et al. (1985) showed that the growth patterns (as characterized by pH, protein, and carbohydrate content of the culture fluid) not only of *A. fumigatus*, but also of *P. notatum* changed during the time of cultivation [81]. Gupta et al. (1999) showed that also in the case of *C. lunata*, the growth time considerably affected the quality and quantity of the harvested raw material [78]. During the initial growth period (days 5–7), the growth medium turned acidic, and the total protein content was high in the fungal mat (Gupta 1999). The later growth period (days 9–11) showed a slight increase in the pH of the growth medium and a further increase in protein and allergen content [78].

Furthermore, *A. alternata* and *C. herbarum* also demonstrated comparable growth patterns [81]. For these species, two phases of growth can be distinguished: phase I that is characterized by a small decrease in the pH, followed by phase II that is characterized by a moderate increase in the pH to more alkaline values [81]. However, *A. alternata* and *C. herbarum* showed less lysis during a longer cultivation when compared to the other fungal species [81]. Moreover, Kroutil et al. (1984) found that even though the total protein content of the mycelial extract prepared from *Alternaria* cultures increased with the growth time, the greatest IgE binding occurred to protein extracts prepared from mycelia harvested after 1 week of cultivation in growth phase I [97]. In our previous study, we analyzed the effect of the cultivation time on the expression of individual *A. alternata* allergens [83]. We saw that the major allergen Alt a 1 was already expressed after 5 days of growth and was also present in comparable amounts in allergen extracts prepared from material harvested after 10, 20 and 30 days, with a slight increase during the longer growth time [83]. The slight increase in the amount of Alt a 1 might be due to the higher number of germinating spores after the longer growth time [83]. We further analyzed the influence of the growth time on the expression of allergenic enolases in different fungal species, including *A. alternata*, *U. chartarum*, *A. fumigatus*, *P. variotii* and *C. herbarum,* and found considerable differences [83]. Under standard conditions, the enolases from *U. chartarum* and *C. herbarum* were only detectable in extracts prepared from older fungal material, whereas the enolases from *A. fumigatus* and *P. variotii* were only expressed in the early stages of growth [83]. Alt a 6, the enolase from *A. alternata*, however, was not detected under the conditions tested [83]. These results show that the impact of the growth time on fungal allergen expression is highly species- and allergen-specific and therefore must be evaluated for each allergenic fungus and each fungal allergen separately [83].

The results presented in these studies suggest that to reach similar allergenic compositions, the growth time must be strictly standardized [96]. Overall, detailed studies of various fungi will be necessary to establish the ideal culture time for each fungus and for each allergen.

While many studies have analyzed the impact of the culture medium, temperature, light and growth time on fungal growth and/or fungal allergen expression, the effects of other factors, such as humidity or atmospheric CO_2_ levels have hardly been investigated [98]. In addition, the impact of different conditions has so far only been studied in a limited number of fungal species and even less is known about the effect of different growth conditions on the expression of specific fungal allergens.

### 3.3. Harvest of Fungal Material

During cultivation, the allergens produced by fungi can occur in the spores, the mycelia, and the growth medium [8,9,10,11,12,13]. Thus, both mycelium and spore-containing material, as well as the growth medium, are possible starting materials for the preparation of allergen extracts [12,13,95]. However, for most fungal allergens, it has not yet been investigated whether they occur in the spores or in the fungal mycelium, or in both, or whether they are readily secreted into the culture medium.

#### 3.3.1. Spores, Mycelia or Growth Medium?

Several studies analyzed the allergen profile of extracts prepared from fungal spores, mycelia or from the growth medium and found, besides biochemical and immunological similarities in the content also considerable differences [8,13,15,52,67,82,84,99,100,101,102,103,104]. In particular, the allergenic properties of different *Aspergillus* and *Alternaria* extracts have been analyzed.

Kauffman et al. (1984) and Reijula et al. (1992) found comparable IgE-binding properties of *A. fumigatus* extracts prepared from spores, mycelia, and growth medium, indicating that relevant *A. fumigatus* allergens are present in spores and mycelia and are released in the culture medium [15,99]. Later analyses that focused on single allergens by Kespohl et al. (2013) showed that during cultivation in liquid medium, the species’ major allergen Asp f 1 was predominantly found in the culture medium filtrate, whereas only low amounts of the allergen were detected in the fungal spores and hyphae [52].

Various studies also analyzed the presence of the major *A. alternata* allergen Alt a 1 in fungal spores, mycelia, and growth medium [8,13,67,84,105,106]. Ibarrola et al. (2004) showed that Alt a 1 was predominantly present in the extract prepared from the growth medium [84]. This was confirmed by another study, where it was shown that Alt a 1 is continuously released into the medium of liquid *A. alternata* cultures, where it accumulates [67,84]. Furthermore, Saenz-de-Santamaria et al. (2006) demonstrated that Alt a 1-homologous allergens, expressed by *S. botryosum*, *Ulocladium botrytis*, *C. lunata* and *Alternaria tenuissima*, were also secreted into the culture medium [40].

When comparing extracts prepared from *Alternaria* spores or mycelium, Aukrust et al. (1985) found that the Alt a 1 concentration was 3 times higher in mycelial than in spore preparations [8]. However, other *Alternaria* allergens were present in higher amounts in the spore preparations [8]. This was confirmed by Paris et al. (1990), who showed that spores contained more allergens than the mycelium, but that the major allergen Alt a 1 was always present in both [13].

Moreover, it has been shown that spore germination plays a crucial role in the secretion of fungal allergens, especially of Asp f 1 and Alt a 1 [84,103,104,105]. Mitakakis et al. (2001) demonstrated the importance of germination for allergen release from *Alternaria* spores as, after germination, a higher number of spores released allergens, including Alt a 1 [103]. The study by Green et al. (2003) confirmed that the germination of *A. alternata* as well as *A. fumigatus* spores increased the amount of detectable allergens and further demonstrated that several other common fungal spores (i.e., *C. herbarum*, *E. nigrum*, *P. chrysogenum*) also released more allergens upon germination [104].

To conclude, many studies indicate that the growth medium represents the best starting material for the preparation of highly allergenic fungal protein extracts. However, it is suggested that reference materials and commercial extracts should rather contain both, spore and mycelial constituents, instead of being prepared from the culture medium only [13,47,107,108]. One reason for this is that even though previous studies found a tendency toward more frequent and pronounced reactions to growth media than to cellular extracts, it is rather difficult to prove that the growth medium contains all the allergens present in the mycelium/spore-containing mat [11,13]. For example, the study by Verma et al. (1994) showed that even though *Fusarium solani* extracts prepared from spores and mycelium contained fewer allergens and in lower concentrations than the extracts prepared from the growth medium, some unique allergens were only detected in the spores and mycelium extracts [109]. However, one has of course to consider that some allergens might only be present in the culture medium due to their fast secretion, such as the major fungal allergen Asp f 1 [9].

#### 3.3.2. Harvest

Depending on the preferred material, manufacturers then harvest the cellular (spores and mycelia) or the extracellular (growth medium) material [12,51]. For this, the fungal mycelia and spores are separated from the culture medium using sieving, filtration, centrifugation, or a combination of these methods [12,55,95]. Then, either the recovered cellular fraction is processed to produce a product suitable for subsequent extraction or the concentrated growth medium is directly used as the extraction fluid [12].

### 3.4. Extraction

Extraction is one of the most critical steps in the manufacturing of an allergen extract [95]. It describes the process of transferring the allergens from the source material into solution [95]. The goal for the manufacturer is to obtain as many and as much of the allergens into solution (major and minor allergens) while minimizing the number of extraneous solutes in the resultant extract [95]. Although the basic concepts of extracting allergens remain the same, each manufacturer has developed variations in the equipment, reagents and processes that are used to produce these extracts [64,95,110]. Therefore, extraction procedures are, in addition to the cultivation conditions, also a source of variation [65]. Studies have shown that the use of different solvents and methods for extraction of the same fungal raw material renders different protein and allergen content, allergen composition and significant differences in the final allergenic potency of the extracts [65]. Factors known to have an impact on the extract quality include the disruption of the cells, the extraction buffer, the ratio of buffer to raw material, the extraction time and temperature and the addition of preservatives, protease inhibitors, or polyvinylpyrrolidone [51,54,111,112]. Extraction conditions have to be determined and optimized for each fungal species [51,64,95].

#### 3.4.1. Disruption Method

Cell disruption is of crucial importance for the preparation of cellular extracts from filamentous fungi due to their robust, chitinous cell wall [95,110]. Thus, mechanical disruption byball mills, food processors, blenders, grinders, homogenizers, and liquid nitrogen crushing, is usually carried out first to break up the fungal material [95,110].

#### 3.4.2. Buffer

Moreover, the buffer as well as the ratio of raw material to buffer can further influence the quality of the protein extract produced [95]. A higher ratio of source material usually results in a higher gross yield of allergens per milliliter but can also lead to the extraction of higher amounts of undesirable material [95]. Concerning extraction fluids, proteins are generally more soluble in saline solutions than in distilled water [95]. Accordingly, most manufacturers use a combination of extracting solutions, including a physiologic saline and/or phosphate buffered saline also with or without preservatives [95].

Results presented in the study by Bouzaine et al. (1989) showed that even though *Cladosporium* can release allergens upon incubation with various chemical solutions, no protein and very little sugar was released when *Cladosporium* spores were incubated in water compared to the extraction in saline solutions [113]. The same effect was seen for *A. alternata*, *A. fumigatus* and *C. herbarum* by Ariaee et al. 2020 [112], thus suggesting that water should not be used as an extraction buffer. In comparison, Paris et al. (1990) demonstrated that the best method for the extraction of *Alternaria* allergens is the breakage of the cells in carbonate buffer containing additives that prevent the precipitation and enzymatic hydrolysis of proteins, e.g., polyvinylpyrrolidone to bind polyphenols as well as phenylmethanesulfonyl fluoride and ethylenediaminetetraacetic acid to inhibit protease activity [13].

#### 3.4.3. Time

Another factor that has a major impact on the extract quality is the extraction time [75,95]. In case of fungal extracts, there exists a tradeoff between maximum allergen recovery and the increased release of nonallergenic substances such as melanin, which is released by highly pigmented spores and phenols, which are known to cause protein precipitations [13,51,112]. Moreover, proteases that might cause the degradation of some allergens might also be extracted during longer extractions times [12,31,110,112,113,114,115,116]. However, the effects of phenols and proteases can be prevented by the addition of phenol-binding components and protease inhibitors [13,110].

The optimal extraction time has already been analyzed for different fungal species [10,95,113,114]. Rijckaert et al. (1980) showed that the main part of the protein-containing material from *Aspergillus penicilloides*, *Aspergillus repens* and *Wallemia sebi* is set free within the first 20 min of extraction, which is interestingyl in accordance with studies that analyzed the release of allergens from pollen [117]. Similar results were obtained by Bouziane et al. (1989), who studied the release of proteins and sugars from *Cladosporium* spores and showed that most of them were released during the initial 2 to 3 h of extraction time [113]. The amount of sugar remained constant until 24 h of extraction, whereas the amount of released protein decreased after 4 h [113]. This correlates with our previous studies, where we showed that some fungal allergens, such as Alt a 1 and different enolases, are immediately released from fungal spores in humid milieu [41,46]. Moreover, Portnoy et al. (1993) found, in the case of *A. alternata,* that each allergen has an optimal extraction time, but they could not determine one specific time point where all relevant allergens were extracted [10].

Despite the results of Rijckaert et al. (1980), who described that most proteins were released within the first 20 min from *Aspergillus penicilloides*, *Aspergillus repens* and *Wallemia sebi* spores, and of Bouziane et al. (1989), who saw that most proteins were released form *Cladosporium* spores during the first 2 to 3 h, commercial fungal allergen preparations are usually obtained after 24 to 48 h of extraction [113,114]. Based on the mentioned studies, a reduction in the extraction time can definitely be recommended.

#### 3.4.4. Temperature

Furthermore, the ideal extraction temperature must be defined. Extraction is typically conducted either at a temperature of 1–5 °C or at controlled room temperature (20–25 °C) [95]. In general, the higher the temperature, the more rapidly allergens can be solubilized; however, it must be considered that many allergens are thermolabile and will be denatured when exposed to high temperatures [95]. Furthermore, at higher temperatures, proteases might also show higher activity.

### 3.5. Purification

As mentioned above, not only fungal allergens are solubilized from the fungal material during the extraction process., Instead, the obtained extracts are complex mixtures of different macromolecules (proteins, glycoproteins and polysaccharides) and low-molecular-weight constituents (pigments and salts) [10,13,51,63,118,119]. They contain allergenic as well as nonallergenic molecules and fungal toxins [13,63,120,121,122,123,124,125]. Therefore, before being used for allergy diagnosis or therapy, fungal allergen extracts must be purified [118,120].

First, once extraction is complete, the insoluble portion of the source material needs to be removed from the liquid extract by centrifugation or filtration [95]. The crude extract can then be further purified by means of centrifugation, filtration, or both, through a series of filters of decreasing porosity [51,95]. Although these are rather simple steps, the conditions and materials used can further determine the final composition of the product [64]. For example, the conditions of centrifugation (e.g., temperature, speed) may reduce the proportion of allergens and the cut-off as well as the nature and composition of filters may interfere in the filtration process, retaining some specific allergens [64]. The supernatant or filtrate might undergo further processing, such as dialysis to remove low-molecular-weight nonallergenic material [51,95]. Moreover, fungal toxins must be removed during the production process through dialysis and defatting procedures [64]. In addition, extracts might be sterilized when used for therapeutic applications by passing them through a sterilizing filter instead of autoclaving, as autoclaving might destroy thermolabile allergens [51,95].

### 3.6. Quality Control

Once a protein extract has been produced and purified for allergy diagnosis or therapy, the extract’s quality must be evaluated [52,64,121]. During quality control, different materials in an extract (i.e., allergens, allergen-derived materials, nonallergenic materials, contaminants) must be analyzed concerning their concentrations, ratios, activity parameters (e.g., allergenic activity, immunogenicity, immunomodulatory activity), shelf-life and stability, and chemical and biological properties related to safety must also be characterized for each of the different components, which is an extremely complex process [121].

Quality control is another part of the extract production process that could serve as a potential source of variability [52,64,121]. For example, it was shown that no single available immunochemical assay is able to properly evaluate the allergenicity of a prepared extract [111]. However, nowadays, methods such as mass spectrometry enable the analysis of single molecules with sufficient accuracy [121]. Mass spectrometry has recently been proposed as a method for the standardization of allergen extracts [121]. However, mass spectrometry can only determine the presence of certain allergen-derived peptides in an extract, but it is not a quantitative method, and it does not allow to draw any conclusions regarding the allergenic or immunogenic properties of the detected molecules [121]. Unfortunately, no method exists that can analyze all important characteristics (physicochemical, structural, immunological properties) of the individual components present in complex mixtures, such as allergen extracts, at the same time [121].

### 3.7. Stability and Storage

One important feature of extract solutions intended to be used for allergy diagnosis, therapy or research is their stability [64,121]. The stability of allergen extracts depends on the type and quantity of allergens, the storage temperature, and the presence of preservatives and other (non-allergenic) materials in the mixture, such as glycerin and protease inhibitors, because any given extract may lose allergenic capacity due to the degradation of specific allergens by proteases [13,31,51,64,112,114,115,116]. Therefore, specific factors affecting stability must be identified to ensure the functional longevity of allergenic extracts [64,112].

Overall, this diversity in the production procedure and the paucity of knowledge about the specific allergens have been the major obstacle for the standardization of commercially available fungal extracts used in clinical practice [12,63]. A summary of the different sources of variability for each production step is shown in Figure 2. However, much remains unknown and generally accepted guidelines are needed to enable extract standardization and, thus, the production of high-quality fungal allergen extracts.

## 4. Standardization of Fungal Allergen Extracts

As discussed in detail above, fungal allergen extracts represent complex mixtures containing, besides the relevant allergenic proteins in unknown concentrations, many non-allergenic proteins, among them also proteases, carbohydrates, and low-molecular-weight substances [52,63,107,118]. As already described in Section 2 and Section 3 and shown in Figure 2, the reasons for the high variability and thus insufficient quality of fungal extracts are manifold [3,23]. The starting material, growth conditions, protein extraction methods and storage conditions all have an influence on the presence and quantity of individual allergens [13,23,123]. All these factors strongly influence the final composition, potency, and stability of allergen preparations and, thusthe outcome of diagnostic tests in vitro and in vivo [52,63].

As described in detail in Section 2, several studies have shown that fungal extracts vary considerably in their protein composition [3,23,43,52,124,125]. Therefore, the standardization of allergen extracts is essential to avoid variations in sensitivity and specificity of allergy diagnosis and to improve the safety and efficacy of immunotherapy [51,52,63,107,118]. The basis of a standardized allergen product are well-controlled reagents and source materials, followed by processing in a robust manufacturing process and quality control based on validated analytical methods [126]. Only when the consistency of quality within batches is assured can efficacy and safety be expected to be similar [107,125,126]. Already in 2004, Robert E. Esch pointed out that a coordinated effort by manufacturers, regulatory authorities, clinicians, and researchers is required [51]. Today, 20 years later, we still have not achieved this.

To comply with the requirements mentioned above, numerous regulations need to be considered and followed by allergen manufacturers, including the European guideline on allergen products from the EMA (2007) and the American guideline on information for an allergenic extract or allergen patch test from the FDA (1999) [126,127].

In Europe, the standardization of allergen products has greatly advanced over the last several years due to the increasing knowledge about individual components that comprise allergenic extracts and the technical progress toward measuring those components, but also due to the increasing pressure from regulatory authorities and the allergist community [128]. From 2007, the Committee for Medicinal Products for Human Use of the European Medicines Agency (EMA) was working on a new guideline for allergen extracts, which was published and came into force in 2009, replacing the previous guideline released in 1996 [64]. This document (EMEA/CHMP/BWP/304831/2007) provides principles and guidance for the manufacturing and quality control of allergen products of biological origin, including allergen extracts from natural source materials and allergens produced through recombinant DNA technology (European Medicines Agency, 2008). A similar guideline on allergenic extracts and allergen patch tests is provided by the FDA, which, however, has not been renewed since 1999.

Both guidelines provide information on the documentation and data that need to be provided by the manufacturers for the marketing of a new allergen product. This includes, for example, a detailed description of the manufacturing process, a precise characterization of the active substance as well as of the finished product, including stability tests, the identification of relevant individual allergens and the measurement of their concentration, and a demonstration of the manufacturer’s capability to obtain batch-to-batch consistency. To be released as a standardized product, an extract must satisfy the criteria of allergenic potency, which is given by regulatory authorities. The potency of allergen extracts is expressed in arbitrary manufacturer-specific units relative to a so-called in-house reference preparation [128]. While the EMA accepts these manufacturer-specific units for standardization, the FDA does not [128]. To facilitate cross-product comparability, the United States require uniform potency-related labeling for each extract for which the Center for Biologics Evaluation and Research (CBER) maintains and distributes reference extracts and serum pools [128]. Thus, while the European system may lead to the standardization of more products, it is not possible to compare extracts between manufacturers [128].

The problem is that most existing regulations exempt already existing, non-standardized products from the more rigorous licensing requirements proving the safety and efficacy that are required for new products [12]. This is also the case for the European guideline, which states that for allergen products that cannot be standardized, for example, due to the unavailability of sufficient patients for potency testing, a range of in vitro methods such as the determination of an antigen profile, protein profile and the content of total protein and individual allergens may be applied for the control of the final product. However, in this case, evidence of allergenic potency is not required, which may be one cause for the marked variability in allergen content [51,52].

In conclusion, even though the standardization of allergen extracts has greatly advanced over the last few years, and national regulatory agencies, especially in the EU and the US, provide guidance documents for the manufacturing and quality control of licensed allergenic extracts, there are, until now, no generally accepted guidelines for the preparation of allergenic fungal extracts [3,12,21,22,23,43,51,128]. The standardization of fungal extracts has been and will continue to be difficult, but it is of highest importance for improved allergy diagnosis, therapy and research is evident [51,63,124,129].

## 5. Conclusions

Fungi represent one of the main causes of respiratory allergy and an alarming increase in sensitization to fungi has been reported. However, today, the problems encountered with the diagnosis of fungal allergy are still far from being solved. The main problem that leads to a general underdiagnosis of fungal allergy, besides the lack of knowledge about fungal species that induce allergic reactions, are the high variability and poor quality of fungal extracts that are commonly used as test solutions for allergy diagnosis. Thus, none of the extracts available has been standardized and approved by regulatory authorities.

Among the manufacturing processes for the production of allergen extracts the processes used to manufacture extracts from fungi show the highest variability. As described in detail in Section 2 and Section 3 and summarized in Figure 2, the reasons for the high variability and thus insufficient quality of fungal extracts are manifold and the starting material, the growth conditions, the protein extraction methods, and the storage conditions all have an influence on the presence and quantity of individual allergens.

Despite the vast variety of studies that analyzed the impact of different production steps on the allergenicity of fungal allergen extracts, much remains unknown. In Figure 3, we summarized, based on the literature that is discussed in this review, possibilities for the reduction of extract variability and for the improvement of extract quality. 

Since fungi represent living organisms that mutate frequently, it is suggested to derive the source material from well-defined, pure seed cultures from established suppliers. However, strain variabilities and the multiplicity of fungal species that are estimated to elicit allergic diseases still represent major challenges. Moreover, it is important to clearly define the cultivation conditions applied to grow the fungi. Studies have shown that the type and concentration of the carbon source in the cultivation medium can especially play a crucial role and that usually a cultivation at 20–37 °C for 2–4 weeks is advisable. Nevertheless, the lack of studies that focus on allergen- and species-specific effects hampers the standardization of fungal cultivation for the preparation of allergen extracts. In addition, also, the process of extracting the allergens from the fungal raw material remains a major source of variability. It is suggested to use a mixture of fungal spores and mycelia for the preparation of the allergen extract instead of the culture medium. This, however, excludes allergens that are readily secreted and are therefore not present in the solid fungal material. Furthermore, based on the literature, the extraction should be performed using saline solutions or distilled water, at temperatures of 1–5 °C or 20–25 °C, for 20 min or until up to 48 h, and after disrupting the cells using standard methods (e.g., bead mill, sonication, mortar). Since extraction processes not only solubilize fungal allergens, but also several non-allergenic components, fungal allergen extracts must be purified before being used for allergy diagnosis or therapy. Even though this seems to be quite a simple step, the methods, conditions, and materials used can further determine the final composition of the product. However, until now, no standardized methods have been proposed. The same is true for one of the final steps of the fungal allergen extract production—the quality control. Quality control is an extremely complex process during which different materials in an extract must be analyzed with respect to their concentrations, ratios, activity parameters, shelf-life and stability, and chemical and biological properties related to safety must be characterized as well. It serves as another source of variability and so far, no standardized method is available that can analyze all the important characteristics of the allergen extract in sufficient sensitivity and specificity. Another important feature of extract solutions intended to be used for allergy diagnosis, therapy or research is their stability. To increase the stability of fungal allergen extracts, preservatives such as glycerin and protease inhibitors should be added to the extracts, which then should ideally be stored at 4 °C or at −20 °C.

This review points to the need for further research in the field of fungal allergology for standardization and for generally accepted guidelines on the preparation of fungal allergen extracts. In particular, the standardization of fungal extracts has been and will continue to be difficult, but it will be crucial for improving allergy diagnosis, therapy, and research.

## Figures and Tables

**Figure 1 jof-09-00957-f001:**
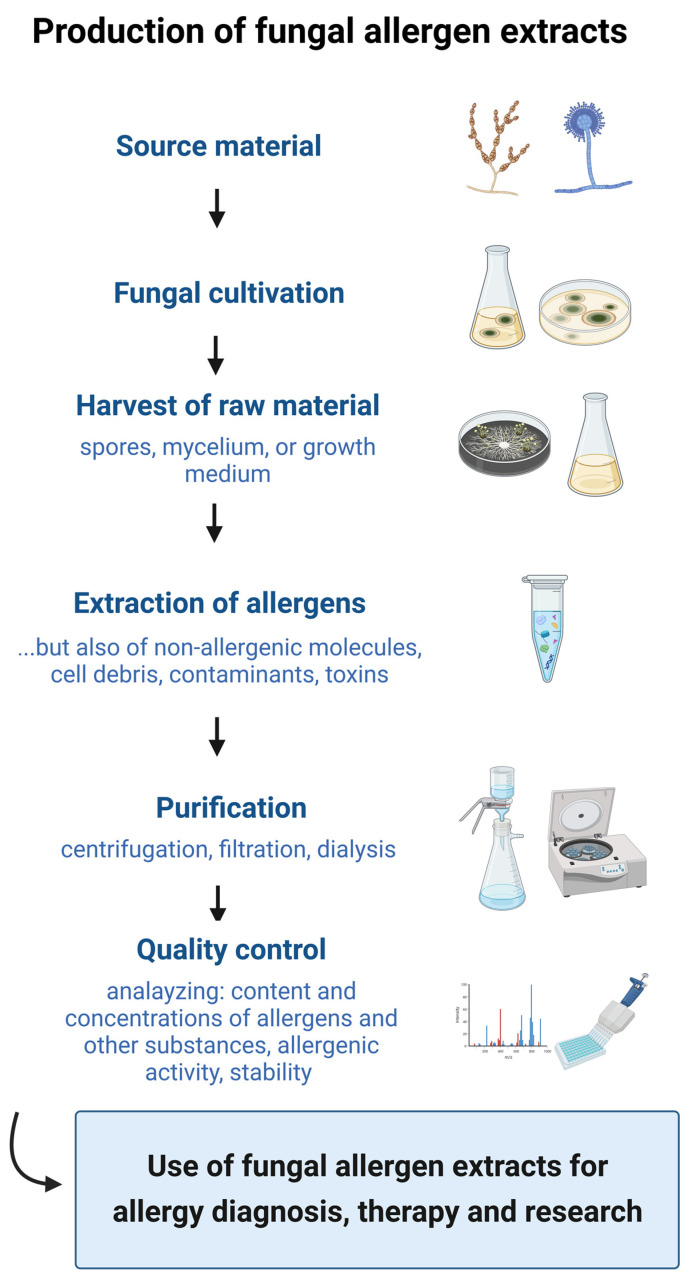
Production steps for the preparation of fungal allergen extracts. The manufacturing process includes the selection of the source material, the fungal cultivation, the harvest of the raw material, the extraction of allergens, the purification, and the quality control (created with BioRender.com).

**Figure 2 jof-09-00957-f002:**
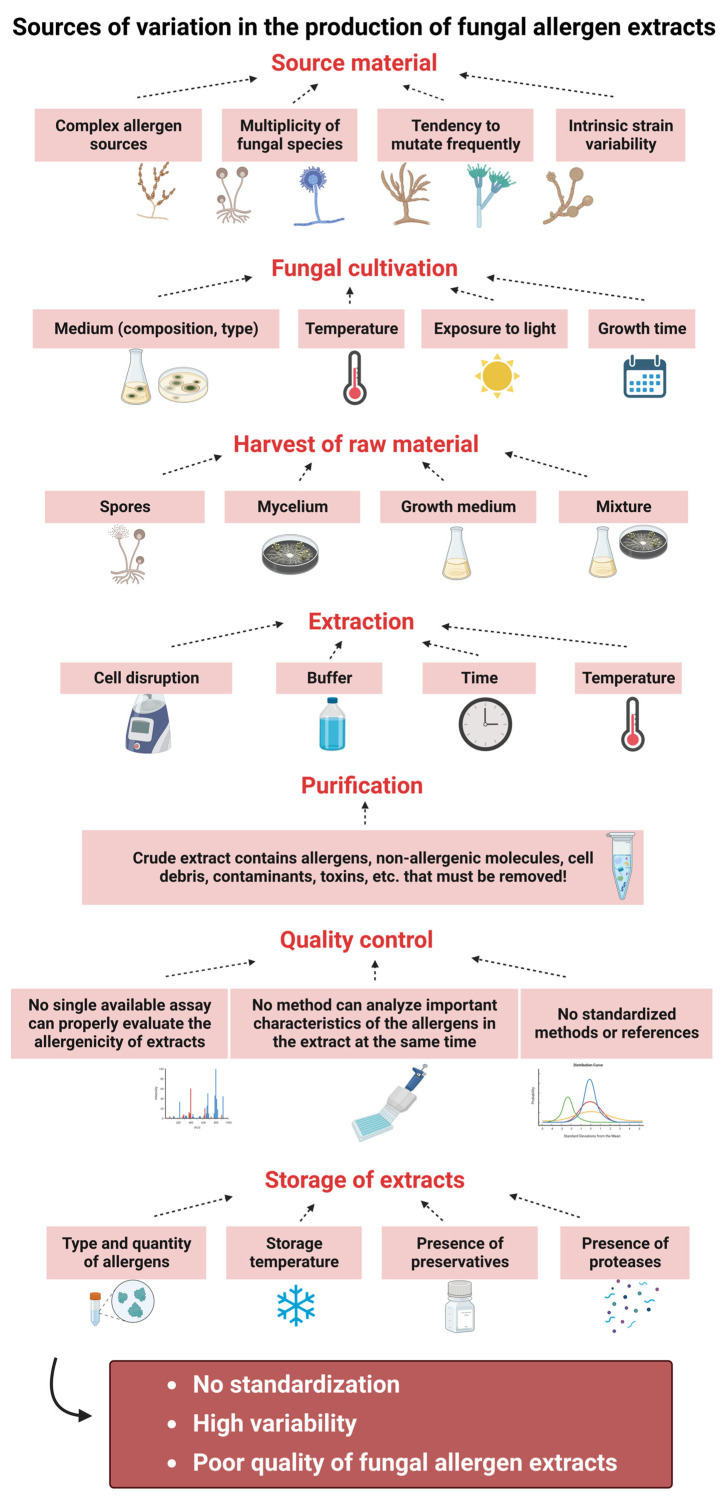
Potential sources of variation in the production of fungal allergen extracts (created with BioRender.com).

**Figure 3 jof-09-00957-f003:**
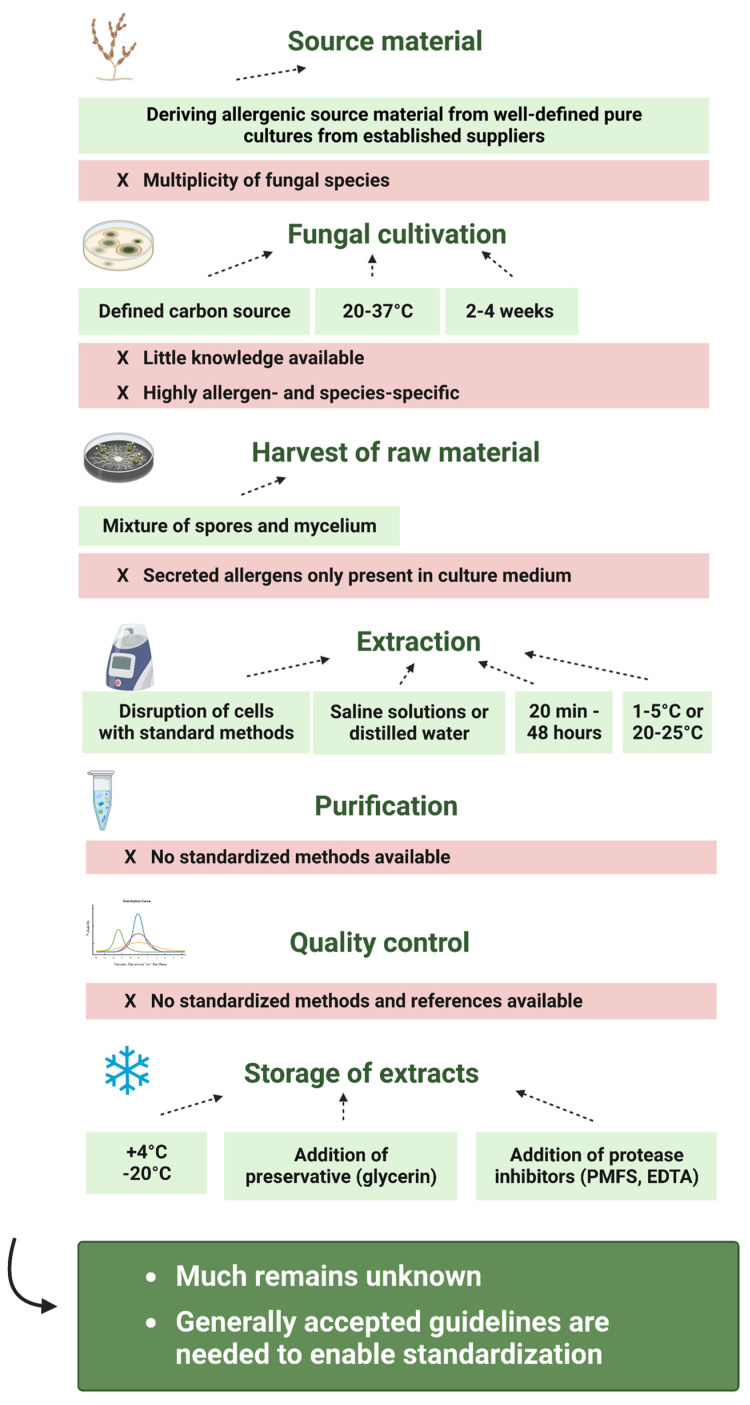
Suggestions on how to obtain high-quality fungal allergen extracts are highlighted in green and the remaining sources of variation on the production of fungal allergen extracts are marked in red (created with BioRender.com).

## Data Availability

Data are contained within the article.

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
