# Peer review of "Problems Encountered Using Fungal Extracts as Test Solutions for Fungal Allergy Diagnosis"

_jof, 2023, doi:10.3390/jof9100957_

Round 1

Reviewer 1 Report

The review “Problems encountered using fungal extracts as test solutions for fungal allergy diagnosis” by Sandra Pfeiffer and Ines Swoboda is focused on the problems encountered in the use of fungi extracts and obtaining a reliable diagnosis. A problem that may also extend to the use of extracts from other sources for allergy diagnosis. The study is well performed and clearly explained, I would only evaluate the figures resolution which is very poor at least in this version, and I woul pay attention to the references in this regard I report an example:

Line 82. The references 6 and 9 used do not refer to the allergens and in general to the subject treated in that context.

Author Response

We would like to thank the reviewer for taking the time to review this manuscript and for the constructive comments that helped us to improve the quality of our manuscript. Please find the detailed responses below and the corresponding corrections highlighted in yellow in the re-submitted files.

Comment 1: The study is well performed and clearly explained, I would only evaluate the figures resolution which is very poor at least in this version.

Reply to comment 1: We thank the reviewer for pointing out the poor quality of the figures. To ensure that the final resolution of the figures is of high quality, we will upload the figures in high resolution separately as jpg-files.

Comment 2: I would pay attention to the references in this regard I report an example:

Line 82. The references 6 and 9 used do not refer to the allergens and in general to the subject treated in that context.

Reply to comment 2: We agree with the reviewer that references 6 (Kurup et al., 2002) and 9 (Barnes 2019) do not refer to the allergens. We therefore removed these references from the mentioned statement in the revised manuscript. In general, we would like to thank the reviewer for pointing our attention to the references. We have checked all references again very carefully and updated them in the revised manuscript.

Reviewer 2 Report

Dear authors,

I read your manuscript concerning the problems encountered using fungal extracts as test solutions for fungal allergy diagnosis. The paper is easy to read and clear, but the reader may find it difficult to search for information in the text and it is strongly recommended to include summary tables in the main text. I report some points to improve the manuscript.

1)     Figure 3, Fungi could be stored in glycerol, not glycerine. Check and clarify in the main text.

2)     Check references style in the main text.  

3)     4Line 208-209, “pure seed”, remove seed. Moreover, there are ATCC and CDC strains as well as the strains deposited in the ECCO and MIRRI collection.

4)     Temperature and CO2 concentration influence fungal growth. Read and cite:

-        Lang-Yona, N., Levin, Y., Dannemiller, K. C., Yarden, O., Peccia, J., & Rudich, Y. (2013). Changes in atmospheric CO2 influence the allergenicity of Aspergillus fumigatus. Global change biology19(8), 2381–2388. https://doi.org/10.1111/gcb.12219

-        Wolf, J., O'Neill, N. R., Rogers, C. A., Muilenberg, M. L., & Ziska, L. H. (2010). Elevated atmospheric carbon dioxide concentrations amplify Alternaria alternata sporulation and total antigen production. Environmental health perspectives118(9), 1223–1228. https://doi.org/10.1289/ehp.0901867

5)     In line 237-244 its explained a simple concept: the use of a poor medium favour the sporulation of the fungus as well as the use of the broth induce the production of vegetative mycelium.

6)     Tables summarizing the results should be reported in the main text.

Minor editing of English language required

Author Response

We would like to thank the reviewer for taking the time to review this manuscript and for the constructive comments that helped us to improve the quality of our manuscript. Please find the detailed responses below and the corresponding corrections highlighted in yellow in the re-submitted files.

Comment 1: Figure 3, Fungi could be stored in glycerol, not glycerine. Check and clarify in the main text.

Reply to comment 1: We thank the reviewer for the comment. We would like to mention that glycerin is the commercial name of glycerol. We decided to use the term “glycerin” instead of “glycerol”, because this term is used in the following references that we cited:

  • E. Esch, “Manufacturing and standardizing fungal allergen products,” Journal of Allergy and Clinical Immunology, vol. 113, no. 2, pp. 210–215, 2004, doi: 10.1016/j.jaci.2003.11.024.
  • E. Esch, “Allergen Source Materials and Quality Control of Allergenic Extracts,” Methods, vol. 13, no. 1, pp. 2–13, Sep. 1997, doi: 10.1006/meth.1997.0491.

Owing to these references, we decided not to change the main text in the revised manuscript.

Comment 2: Check references style in the main text.

Reply to comment 2: We just followed the reference style mentioned in the guidelines of the journal.

Comment 3: Line 208-209, “pure seed”, remove seed. Moreover, there are ATCC and CDC strains as well as the strains deposited in the ECCO and MIRRI collection.

Reply to comment 3: We thank the reviewer for the suggestion to remove “seed”. We rewrote the sentence in the revised manuscript (see Lines 206-211). In case of the suppliers of microbial material (ATCC, CDC, ECCO, MIRRI), we kept the statement as it was, since we did not intend to provide a full list of suppliers, but only wanted to mention some examples of suppliers.

 Comment 4: Temperature and CO2 concentration influence fungal growth. Read and cite:

  • Lang-Yona, N., Levin, Y., Dannemiller, K. C., Yarden, O., Peccia, J., & Rudich, Y. (2013). Changes in atmospheric CO2 influence the allergenicity of Aspergillus fumigatus. Global change biology19(8), 2381–2388. https://doi.org/10.1111/gcb.12219
  • Wolf, J., O'Neill, N. R., Rogers, C. A., Muilenberg, M. L., & Ziska, L. H. (2010). Elevated atmospheric carbon dioxide concentrations amplify Alternaria alternata sporulation and total antigen production. Environmental health perspectives118(9), 1223–1228. https://doi.org/10.1289/ehp.0901867

Reply to comment 4: We appreciate the reviewer’s comment and added the information presented in the studies according to the suggestion of the reviewer in the revised manuscript in Lines 404-407.

Comment 5: In line 237-244 its explained a simple concept: the use of a poor medium favour the sporulation of the fungus as well as the use of the broth induce the production of vegetative mycelium.

Reply to comment 5: In the paragraph mentioned by the reviewer, the impact of solid as compared to liquid medium on fungal allergen expression is described. However, the impact of poor as compared to rich medium on fungal growth and/or fungal allergen expression is not discussed in this paragraph. This is also not part of the studies cited in this paragraph. Therefore, we decided to keep this paragraph unchanged in the revised manuscript.

Comment 6: Tables summarizing the results should be reported in the main text.

Reply to comment 6: We appreciate the comment of the reviewer stating that tables, which summarize the reported results, would be beneficial for the quality of the manuscript. We tried to include such tables, however, due to the great variability of the cited studies, we were not able to summarize the results in the form of easily comprehendible tables. Thus, we decided against including any tables in the revised manuscript, because we had the feeling that tables might lead to misleading and/or confusing conclusions. However, we kept the summarizing figures in the revised manuscript.

Comments on the Quality of English Language: Minor editing of English language required.

Reply to comment: We thank the reviewer for this remark and asked a native speaker to proofread the manuscript.